# DYNAMICS OF LEARNING WHEN LEARNING DYNAMICS USING NEURAL NETWORKS

## ABSTRACT

When neural networks are trained from data to model the dynamics of physical systems, they encounter a persistent challenge: the long-time dynamics they produce are often unphysical or unstable. We analyze the origin of such unphysical instabilities when learning linear dynamical systems, focusing on the learning dynamics of gradient descent. We make several analytical findings, which empirical observations suggest extend to nonlinear dynamical systems. First, the rate of convergence of the learning dynamics of gradient descent is uneven and depends on the distribution of energy in the data. As a special case, in directions in which the data have no energy, the true dynamics of the physical system cannot be learned. High dimensionality also inhibits learning. Second, in the unlearnable directions, the model dynamics produced by the neural network depend on the weight initialization, and common weight initialization schemes can produce unstable model dynamics. Third, injecting synthetic noise into the data adds damping to the learning dynamics and can stabilize the learned model dynamics, though doing so undesirably biases the learned model dynamics. For each contributor to unphysical instability, we suggest mitigative strategies. We also highlight important differences between learning discrete-time and continuous-time dynamics, and discuss extensions to nonlinear systems. Numerical experiments on a nonlinear dynamical system show where the theory succeeds, where it fails, and where future research should be directed.

## 1 INTRODUCTION

The ability to accurately and efficiently predict the behavior of a dynamical system is fundamental to science and engineering. As data have become increasingly available, machine learning has been leveraged to learn models that can accurately and efficiently simulate the dynamics of physical systems, both in cases when the true dynamics are unknown, or when they are known but are expensive to compute using traditional numerical methods (Pathak et al., 2018; Vlachas et al., 2018; Linot & Graham, 2020; Karniadakis et al., 2021; Pfaff et al., 2021; Linot & Graham, 2022; Srinivasan et al., 2022; Vlachas et al., 2022; Chen et al., 2022; Floryan & Graham, 2022). Weather forecasting offers a prime example of this new approach, where decades of high-quality data (Hersbach et al., 2020) have been used to develop deep learning-based forecasters with accuracy comparable to traditional numerical weather prediction while offering significant speedups (Pathak et al., 2022; Bi et al., 2023; Lam et al., 2023).

While accurate at short times, neural network-based physics simulators can produce unphysical or unstable predictions at long times (Vlachas et al., 2018; Pfaff et al., 2021; Linot & Graham, 2022; Stachenfeld et al., 2022; Keisler, 2022; Chattopadhyay & Hassanzadeh, 2023). Although certain stabilization tricks have been identified, it is unclear when they may be employed, why they succeed, and when they may fail. There is currently no quantitative, theoretical understanding of why neural network-based physics simulators fail at long times. Such a theory would not only reveal the root cause of failure, but, crucially, it would also provide tremendous insight in the development of principled mitigative strategies.

Here, we develop a theory for the learning dynamics of neural networks tasked with emulating a dynamical system. By analyzing the learning dynamics, we identify probable culprits for the origin of the unphysical instabilities that mar the long-time predictions of neural networks. To render the

problem analytically tractable, we consider linear dynamical systems, which can be perfectly emulated by a linear single-layer neural network. This provides an important starting point for understanding the learning dynamics of deep neural networks tasked with emulating nonlinear dynamical systems, and is in keeping with approaches used for prior developments in our understanding of the learning dynamics of neural networks (Baldi & Hornik, 1989; Fukumizu, 1998; Saxe et al., 2014; 2019; Lampinen & Ganguli, 2019; Zhang et al., 2025). Our analysis considers a prototypical neural network training scheme, consisting of the mean squared error loss function, Glorot weight initialization (Glorot & Bengio, 2010), and gradient descent to update the weights of the neural network. We analyze discrete- and continuous-time dynamical systems in turn, highlighting important differences that arise between these two classes of dynamical systems. Our analysis focuses on the effects of energy distribution, weight initialization, and noise on the learning dynamics. Along the way, we propose mitigative strategies and rationalize stabilization tricks found in the literature. Finally, we perform numerical experiments on the Kuramoto-Sivashinsky system, a prototypical nonlinear dynamical system.

## 2 DISCRETE-TIME DYNAMICAL SYSTEMS

Consider the discrete-time dynamical system

$$x_{i+1} = f(x_i), \qquad x_i \in \mathbb{R}^n. \tag{1}$$

Suppose that the dynamics $f$ is not known explicitly, and we want to learn an accurate model $\hat{f} : \mathbb{R}^n \to \mathbb{R}^n$ from a dataset of pairs of snapshots, $\{(x_i, y_i)\}_{i=1}^m$, that are generated by the dynamical system. In our notation, $x_i$ represents the present state of the dynamical system, and $y_i$ the future state, so that

$$y_i = f(x_i), \qquad i = 1, \dots, m. \tag{2}$$

We begin by analyzing a model consisting of a linear single-layer neural network, $\hat{f} = \hat{A} \in \mathbb{R}^{n \times n}$. Assembling the data into the data matrices $X = [x_1 \ \cdots \ x_m]$ and $Y = [y_1 \ \cdots \ y_m]$, which are related by $Y = f(X)$ applied columnwise, we seek the $\hat{A}$ that minimizes the mean squared error loss function $L$,

$$\min_{\hat{A}} \quad \frac{1}{2mn} \|Y - \hat{A}X\|_F^2. \tag{3}$$

Here, $\| \cdot \|_F$ is the Frobenius norm, the factor of $\frac{1}{2}$ is for convenience, and the factor of $\frac{1}{mn}$ mirrors how this loss function is typically implemented in machine learning software packages. The minimizer may not be unique. The least-squares/minimum-norm solution is $\hat{A} = YX^+$, where $X^+$ is the pseudoinverse of $X$. However, what is of interest here is what the gradient descent algorithm produces.

The gradient of $L$ with respect to $\hat{A}$ is

$$\nabla_{\hat{A}} L = -\frac{1}{mn}(Y - \hat{A}X)X^T. \tag{4}$$

In the limit of small learning rate, gradient descent leads to continuous-time gradient flow dynamics

$$\frac{\mathrm{d}}{\mathrm{d}\tau} \hat{A} = -\nabla_{\hat{A}} L, \tag{5}$$

with $\tau$ being a pseudo-time variable. The learning rate is absorbed by $\tau$.

The solution to this matrix ordinary differential equation is

$$\hat{A}(\tau) = \hat{A}(0) \exp\left(-\frac{1}{mn}XX^T\tau\right) + YX^+ \left[I - \exp\left(-\frac{1}{mn}XX^T\tau\right)\right], \tag{6}$$

where $\hat{A}(0)$ is the initialization of the weight matrix. This solution provides the entire time course of learning of the weight matrix. No assumptions have been made in arriving at this solution. The homogeneous solution (the first term) gives an exponential decay of the weight initialization, and the particular solution (the second term) gives an exponential approach toward the least-squares/minimum-norm solution. In the case that $r = \text{rank}(X) = n$, the learned weight matrix $\hat{A}$ converges to the the least-squares/minimum-norm solution in the limit $\tau \to \infty$.

The more interesting case is when $r < n$. This occurs when the data have low-dimensional structure, or when the number of samples $m < n$. The former is common for data from physical systems, and the latter occurs for high-dimensional dynamical systems. In this case, $XX^T$ is only positive *semi*definite, and the matrix exponentials do not decay to zero. No matter how long learning proceeds for, the initialization of the weight matrix persists in the learned weight matrix. In other words, the learned weight matrix depends on its initialization.

A coordinate change proves to be insightful in understanding precisely how the initialization affects the weight matrix. Let $X = U\Sigma V^T$ be the singular value decomposition of $X$, and $\hat{\tilde{A}} = U^T \hat{A} U$ the weight matrix in the basis of the left singular vectors. Then the gradient flow dynamics can be re-written as

$$\frac{\mathrm{d}}{\mathrm{d}\tau}\hat{\tilde{A}} = \frac{1}{mn}(\tilde{Y}V\Sigma^T - \hat{\tilde{A}}\Sigma\Sigma^T), \tag{7}$$

where $\tilde{Y} = U^T Y$ gives the coordinates of the data vectors contained in $Y$ in the basis of the left singular vectors. In these coordinates, the solution is

$$\hat{\tilde{A}}(\tau) = \hat{\tilde{A}}(0)\exp\left(-\frac{1}{mn}\Sigma\Sigma^T\tau\right) + \tilde{Y}V\Sigma^+\left[I - \exp\left(-\frac{1}{mn}\Sigma\Sigma^T\tau\right)\right]. \tag{8}$$

Since $\Sigma$ is a diagonal matrix whose first $r$ entries are positive (and the rest zero), we see that, in the limit $\tau \to \infty$, the first $r$ columns of $\hat{\tilde{A}}$ converge to the first $r$ columns of $\tilde{Y}V\Sigma^+$ (which is a similarity transformation of the least-squares/minimum-norm solution $YX^+$, and whose last $n - r$ columns are zeros), while the last $n - r$ columns of $\hat{\tilde{A}}$ remain at all times equal to their initialization.

For further insight, we specialize to the case of linear dynamical systems, that is, when the true dynamics $f = A \in \mathbb{R}^{n \times n}$. Making the substitution $Y = AX$, the solution to the learning dynamics is

$$\hat{\tilde{A}}(\tau) = \tilde{A} + [\hat{\tilde{A}}(0) - \tilde{A}]\exp\left(-\frac{1}{mn}\Sigma\Sigma^T\tau\right). \tag{9}$$

Three points of interest arise. First, columns of $\hat{\tilde{A}}$ corresponding to non-zero singular values converge to the corresponding columns of $\tilde{A}$—that is, the true dynamics—while those corresponding to zero singular values remain equal to their initial values. In other words, the true dynamics are not learnable in directions in which the data have no energy (equivalently, in directions in which the data have no variance, though we prefer to say "energy" in order to maintain the connection to physical systems). Second, the rate of convergence depends on how energy is distributed in the data. In directions in which the data have low energy, convergence to the true dynamics is slower. It is, therefore, more difficult to learn the dynamics of low-energy modes. The convergence rates can be made uniform by normalizing the data so that they have equal energy in all directions (that is, by whitening the data). Additionally, by noting that $\frac{1}{m}\Sigma\Sigma^T$ is the diagonalized covariance matrix of the data $X$, we see that the rate of learning has a factor of $\frac{1}{n}$, $n$ being the dimension of the state of our dynamical system. Therefore, learning is slower for high-dimensional systems. Third, the initialization of the weight matrix will impact the learned dynamics in the unlearnable directions, or in all directions if gradient descent is stopped short of convergence (which is typical).

To illuminate the impact of the initialization on the learned dynamics, we suppose the data are in a set that is invariant under the dynamics of our system. This is generally true for the large class of physical systems with energy dissipation (due to the presence of friction, for example), for which the long-time dynamics approach an invariant manifold (Hopf, 1948; Foias et al., 1988; Temam & Wang, 1994; Doering & Gibbon, 1995). For a linear dynamical system, the data live in an invariant subspace, and the true dynamics then take the form

$$\tilde{A} = \begin{bmatrix} \tilde{A}_{11} & \tilde{A}_{12} \\ 0 & \tilde{A}_{22} \end{bmatrix}, \tag{10}$$

where the upper-left submatrix gives the dynamics in the invariant subspace containing the data, and the other columns correspond to the unlearnable directions. As $\tau \to \infty$, the learned dynamics will be

$$\hat{\tilde{A}} = \begin{bmatrix} \tilde{A}_{11} & \hat{\tilde{A}}_{12}(0) \\ 0 & \hat{\tilde{A}}_{22}(0) \end{bmatrix}. \tag{11}$$

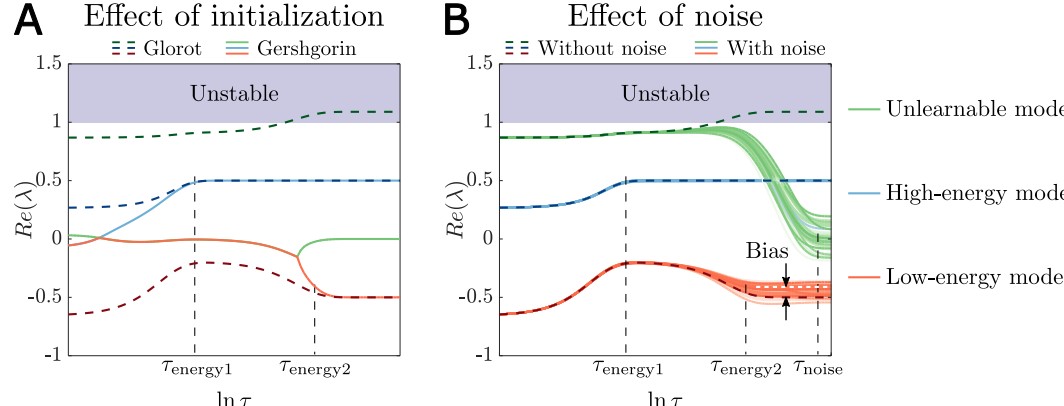

Figure 1: Learning dynamics for a three-dimensional discrete-time system. The real parts of the eigenvalues of the learned dynamical system are shown as learning progresses. The true eigenvalues are 0.5 (high-energy mode), $-0.5$ (low-energy mode), and $-0.5$ (unlearnable mode). The high-energy mode converges more quickly (blue; in time $\tau_{\text{energy1}} \sim 1/\sigma_1^2$) than the low-energy mode (red; in time $\tau_{\text{energy2}} \sim 1/\sigma_2^2$). (a) The initialization scheme based on Gershgorin's circle theorem stabilizes the unlearnable and otherwise unstable mode (green), and the dynamics of the other modes are learned correctly. (b) Noise stabilizes the unlearnable and otherwise unstable mode (green; in time $\tau_{\text{noise}} \sim 1/\sigma^2$), but biases the learnable dynamics.

The eigenvalues of the learned dynamics are the union of the eigenvalues of $\tilde{A}_{11}$ and the eigenvalues of $\hat{\tilde{A}}_{22}(0)$. If rank$(X) = r$ and $\hat{A}(0)$ is initialized using the typical Glorot normal or uniform initializers, then as $n \to \infty$, the distribution of the eigenvalues of $\hat{\tilde{A}}_{22}(0)$ converges almost surely to the uniform distribution on the disk of radius $\sqrt{\frac{n-r}{n}}$ centered at the origin (Tao et al., 2010). For finite $n$, however, the eigenvalues can lie outside the unit circle, as shown in Appendix A, leaving the potential for unstable dynamics.

This point bears emphasizing. In physical systems with dissipation, the state of the system will quickly approach an invariant set. When we collect measurements of the state, those measurements will generally not include information outside of the invariant set due to its strong stability. Without such information, it is impossible to learn the dynamics outside of the invariant set; the learned dynamics in that region will depend on the initialization of the weights. Ironically, it is the strong stability of the true dynamics that can lead to unstable learned dynamics since there is no information available in the data to erase the memory of the potentially unstable weight initialization.

Two remedies are clear. One is to restrict the learned model to remain in the invariant set. For linear systems, this amounts to projecting the system onto the subspace with non-zero singular values. The second remedy is to ensure that the initialization gives stable dynamics, which requires developing new weight initialization schemes. Gershgorin's circle theorem provides one way to create a mostly random matrix whose eigenvalues are guaranteed to lie inside the unit circle, thereby giving stable dynamics. For example, by drawing the entries of $\hat{A}(0)$ from the uniform distribution on $\frac{1}{n-1}[-1, 1]$ and setting the diagonal entries equal to zero, the eigenvalues of $\hat{A}(0)$ are bounded by the unit circle. This bound is sharp (e.g., if all off-diagonal entries are equal to $\frac{1}{n-1}$ then 1 is an eigenvalue) but almost surely conservative, as demonstrated in Appendix A. The distribution of eigenvalues can be expanded (but still bounded by the unit circle) by re-normalizing each row (or column) of $\hat{A}(0)$ so that the row (column) sums of the absolute values of the entries in each row (column) are equal to 1.

The stability of the initialization scheme based on Gershgorin's circle theorem is shown in Figure 1a for a three-dimensional system. More details of the system are available in Appendix B. The dynamics of the learnable modes are learned correctly, while those of the unlearnable mode are made stable. In contrast, Glorot normal initialization leads to incorrect unstable learned dynamics for the unlearnable mode.

## 2.1 NOISY DATA

Curiously, it has been observed that injecting synthetic noise into the data during training can help stabilize neural network-based physics simulators (Vlachas et al., 2018; 2020; Sanchez-Gonzalez et al., 2020; Pfaff et al., 2021; Stachenfeld et al., 2022; Su et al., 2022). To understand this empirical observation, we consider how the addition of noise to the data affects the learning dynamics.

Suppose our measurements are noisy, so that the data matrices are $X + N_x$ instead of $X$, and $Y + N_y$ instead of $Y$. $N_x$ and $N_y$ are random matrices of noise, with all entries assumed to be independent random variables with zero mean and variance $\sigma^2$, and independent of the noise-free data. It is likely that many columns of $N_y$ are the same as columns of $N_x$, but shifted over one column in location, since data are often gathered from long trajectories that provide many data. We proceed with the case where the true dynamical system is linear. With noisy data, the gradient of the loss function with respect to $\hat{A}$ is

$$\nabla_{\hat{A}} L = -\frac{1}{mn}[(A - \hat{A})(X + N_x) + N_y - A N_x](X + N_x)^T. \tag{12}$$

The learning dynamics yield the same solution as in equation 8, but with the data matrices, the singular vectors and values, and the associated coordinate change replaced by their noisy counterparts. With the addition of noise, we generally have $\mathrm{rank}(X + N_x) = \min(m, n) \geq r = \mathrm{rank}(X)$, so the learned weight matrix will be less affected by its initialization than in the noise-free case. Although the solution is clean, it obscures the precise effect of the noise.

The effect of noise can be seen explicitly by using the singular value decomposition of the noise-free data $X$. In the basis of the left singular vectors of $X$, the learning dynamics give

$$\hat{\tilde{A}}(\tau) = \hat{\tilde{A}}(0) \exp\left[-\frac{1}{mn}(\Sigma V^T + \tilde{N}_x)(\Sigma V^T + \tilde{N}_x)^T \tau\right]$$

$$+ \tilde{A} P \left\{ I - \exp\left[-\frac{1}{mn}(\Sigma V^T + \tilde{N}_x)(\Sigma V^T + \tilde{N}_x)^T \tau\right]\right\}$$

$$+ (\tilde{N}_y - \tilde{A}\tilde{N}_x)(V\Sigma^T + \tilde{N}_x^T)\left[(\Sigma V^T + \tilde{N}_x)(\Sigma V^T + \tilde{N}_x)^T\right]^+$$

$$\times \left\{ I - \exp\left[-\frac{1}{mn}(\Sigma V^T + \tilde{N}_x)(\Sigma V^T + \tilde{N}_x)^T \tau\right]\right\}, \tag{13}$$

where $\tilde{N}_x = U^T N_x$, $\tilde{N}_y = U^T N_y$, and $P$ is the orthogonal projection onto the subspace spanned by the noisy data vectors comprising $X + N_x$. If $\mathrm{rank}(X + N_x) = n$, which is generally true for $m \geq n$, then $P$ is the identity and the pseudoinverse above becomes the inverse. In this case, in the limit $\tau \to \infty$,

$$\hat{\tilde{A}} = \tilde{A} + (\tilde{N}_y - \tilde{A}\tilde{N}_x)(V\Sigma^T + \tilde{N}_x^T)\left[(\Sigma V^T + \tilde{N}_x)(\Sigma V^T + \tilde{N}_x)^T\right]^{-1}, \tag{14}$$

which converges in probability to

$$\hat{\tilde{A}} = \tilde{A}[I - m\sigma^2(\Sigma\Sigma^T + m\sigma^2 I)^{-1}]. \tag{15}$$

Rewriting the above expression provides clarity:

$$\hat{\tilde{A}} = \tilde{A} \begin{bmatrix} \frac{\sigma_1^2}{\sigma_1^2 + m\sigma^2} & & & & \\ & \ddots & & & \\ & & \frac{\sigma_r^2}{\sigma_r^2 + m\sigma^2} & & \\ & & & 0 & \\ & & & & \ddots \end{bmatrix}. \tag{16}$$

We see that when the data are noisy, $\hat{\tilde{A}}$ is a biased version of $\tilde{A}$. In particular, columns of $\hat{\tilde{A}}$ corresponding to non-zero singular values converge to the corresponding columns of $\tilde{A}$, but biased by a multiplicative factor $\sigma_i^2/(\sigma_i^2 + m\sigma^2) \leq 1$, where $\sigma_i$ is the $i^{\text{th}}$ singular value of $X$. This bias

factor can be written as $SNR_i/(1 + SNR_i)$, where $SNR_i = \sigma_i^2/m\sigma^2$ is the signal-to-noise ratio in the direction of the $i^{\text{th}}$ singular vector. This is analogous to the attenuation bias in ordinary least squares regression due to classical errors-in-variables (Wooldridge, 2010, Ch. 4.4.2). In contrast, without noise, these columns of $\hat{\tilde{A}}$ converge to the true columns of $\tilde{A}$. The columns corresponding to zero singular values converge to zero, while they stayed equal to their initialization when there was no noise. In equation 13, we see that the noise adds damping to the learning dynamics, erasing the memory of the initialization in the unlearnable directions and replacing it with dynamics that are strongly stable. This is a highly desirable effect since, as previously explained, in physical systems with dissipation, the stabilizing effect of dissipation is what makes those directions unlearnable.

How fast is the memory of the weight initialization erased by the noise? The corresponding eigenvalues of the gradient flow dynamics are, in expectation, equal to $-\sigma^2/n$. The rate of convergence depends on the strength of the noise and the dimension of the system, with weak noise and a large dimension of the system leading to slow convergence. Note that if $n > m$, the initialization of the weight matrix will affect the solution in the limit $\tau \to \infty$. That is because, in this case, the range of the noise matrix is smaller than $\mathbb{R}^n$. One way to mitigate this issue could be to vary the noise that is added to the data so that it spans different subspaces of $\mathbb{R}^n$ at each iteration of the learning process. Since noise adds damping to the learning dynamics, varying the noise could have the effect of varying the directions in which damping is added to the learning dynamics, the net effect being that there is additional damping in all directions throughout the learning dynamics, causing the memory of the initialization to be erased. Whether this is indeed what happens requires the analysis of a matrix stochastic differential equation in which the noise appears nonlinearly, which we leave for future work.

There is a tradeoff between the desirable and undesirable effects of noise: noise stabilizes a learned physics simulator, with stronger noise stabilizing the learned system more quickly, but stronger noise also leads to greater bias. This is illustrated in Figure 1b for the three-dimensional system described in Appendix B, along with the effect of energy distribution. It may be possible to obtain stability while avoiding bias by selectively applying noise only in the unlearnable/zero-energy directions.

## 3 CONTINUOUS-TIME DYNAMICAL SYSTEMS

Important differences arise in continuous-time dynamics. Consider the linear continuous-time dynamical system

$$\frac{\mathrm{d}}{\mathrm{d}t}x = Ax, \qquad x(t) \in \mathbb{R}^n, A \in \mathbb{R}^{n \times n}. \tag{17}$$

As in the discrete-time case, suppose that $A$ is not known explicitly, and we want to learn an accurate model $\hat{A} \in \mathbb{R}^{n \times n}$ from a dataset of pairs of snapshots separated by a time $\Delta t$, $\{(x(t_i), x(t_i + \Delta t))\}_{i=1}^m = \{(x_i, y_i)\}_{i=1}^m$, that are generated by the dynamical system, so that

$$y_i = e^{A\Delta t}x_i, \qquad i = 1, \ldots, m. \tag{18}$$

This learning problem is the same as that of neural ODEs (Chen et al., 2018). Assembling the data into data matrices as before, we seek the $\hat{A}$ that minimizes the mean squared error $L$,

$$\min_{\hat{A}} \quad \frac{1}{2mn}\|Y - e^{\hat{A}\Delta t}X\|_F^2. \tag{19}$$

Due to the presence of the matrix exponential, there is no simple expression for the gradient of $L$ with respect to $\hat{A}$, and the learning dynamics are highly nonlinear in $\hat{A}$. To gain insight, we expand the matrix exponential to $\mathcal{O}(\Delta t)$—equivalent to solving the dynamics using the forward Euler method, making the learning problem equivalent to learning a residual network (scaled by $\Delta t$) (He et al., 2016). Under this approximation, the gradient of the loss function is

$$\nabla_{\hat{A}}L = -\frac{\Delta t^2}{mn}(A - \hat{A})XX^T, \tag{20}$$

equal to that in the discrete-time setting but scaled by a factor $\Delta t^2$. The conclusions made in the discrete-time setting therefore extend to the continuous-time setting, but for the following exceptions.

First, the factor of $\Delta t^2$ in the gradient changes the rate of convergence of the learning dynamics. Small $\Delta t$ could lead to very slow convergence.

Second, for continuous-time dynamics, an instability arises when the real part of any eigenvalue of $\hat{A}$ is positive. When $\hat{A}(0)$ is created using the Glorot initializer, each of its eigenvalues will have a positive real part with probability $\frac{1}{2}$. The continuous-time problem is therefore more susceptible to instability. Gershgorin's circle theorem can again be used to create a stable mostly random matrix by having the Gershgorin disks lie in the left half of the complex plane (or within the stable region of the numerical integrator begin used; for example, if using a residual network, the Gershgorin disks should be made to lie within the stability boundary of the forward Euler method).

Finally, the effects of measurement noise differ in the details. With noisy data, the gradient is

$$\nabla_{\hat{A}} L = -\frac{\Delta t}{mn} \left[ \Delta t(A - \hat{A})(X + N_x) + N_y - (I + A\Delta t)N_x \right] (X + N_x)^T. \tag{21}$$

In the basis of the left singular vectors of the noise-free data $X$, the learning dynamics give

$$\hat{\tilde{A}}(\tau) = \hat{\tilde{A}}(0) \exp\left[ -\frac{\Delta t^2}{mn}(\Sigma V^T + \tilde{N}_x)(\Sigma V^T + \tilde{N}_x)^T \tau \right]$$

$$+ \left[ \tilde{A}P + \frac{1}{\Delta t}(P - I) \right] \left\{ I - \exp\left[ -\frac{\Delta t^2}{mn}(\Sigma V^T + \tilde{N}_x)(\Sigma V^T + \tilde{N}_x)^T \tau \right] \right\}$$

$$+ \frac{1}{\Delta t}[\tilde{N}_y - (I + \tilde{A}\Delta t)\tilde{N}_x](V\Sigma^T + \tilde{N}_x^T)\left[ (\Sigma V^T + \tilde{N}_x)(\Sigma V^T + \tilde{N}_x)^T \right]^+$$

$$\times \left\{ I - \exp\left[ -\frac{\Delta t^2}{mn}(\Sigma V^T + \tilde{N}_x)(\Sigma V^T + \tilde{N}_x)^T \tau \right] \right\}, \tag{22}$$

where $P$ is as in equation 13. If $\text{rank}(X + N_x) = n$, then $P$ is the identity and the pseudoinverse above becomes the inverse. In this case, In the limit $\tau \to \infty$,

$$\hat{\tilde{A}} = \tilde{A} + \frac{1}{\Delta t}[\tilde{N}_y - (I + \tilde{A}\Delta t)\tilde{N}_x](V\Sigma^T + \tilde{N}_x^T)\left[ (\Sigma V^T + \tilde{N}_x)(\Sigma V^T + \tilde{N}_x)^T \right]^{-1}, \tag{23}$$

which converges in probability to

$$\hat{\tilde{A}} = \tilde{A}[I - m\sigma^2(\Sigma\Sigma^T + m\sigma^2 I)^{-1}] - \frac{m\sigma^2}{\Delta t}(\Sigma\Sigma^T + m\sigma^2 I)^{-1}. \tag{24}$$

Rewriting the above expression provides clarity:

$$\hat{\tilde{A}} = \tilde{A} \begin{bmatrix} \frac{\sigma_1^2}{\sigma_1^2 + m\sigma^2} & & & & \\ & \ddots & & & \\ & & \frac{\sigma_r^2}{\sigma_r^2 + m\sigma^2} & & \\ & & & 0 & \\ & & & & \ddots \end{bmatrix} - \frac{1}{\Delta t} \begin{bmatrix} \frac{m\sigma^2}{\sigma_1^2 + m\sigma^2} & & & & \\ & \ddots & & & \\ & & \frac{m\sigma^2}{\sigma_r^2 + m\sigma^2} & & \\ & & & 1 & \\ & & & & \ddots \end{bmatrix}. \tag{25}$$

As in the discrete-time setting, noise creates a multiplicative bias factor. Additionally, there is an additive bias that can be substantial for small $\Delta t$. The columns corresponding to zero singular values converge to columns whose only non-zero entries are along the diagonal and are equal to $-1/\Delta t$. Noise again erases the memory of the weight initialization in the unlearnable directions and replaces it with stable dynamics, with smaller $\Delta t$ leading to more stable dynamics. In addition to the tradeoffs noted in the discrete-time setting, in the continuous-time setting $\Delta t$ also has a tradeoff: smaller $\Delta t$ creates more stable dynamics but greater bias.

## 4 NUMERICAL EXPERIMENTS ON THE KURAMOTO-SIVASHINSKY SYSTEM

The theoretical results of the previous sections were produced in idealized linear settings. Here, we test some of the results for deep, nonlinear neural networks tasked with emulating the dynamics

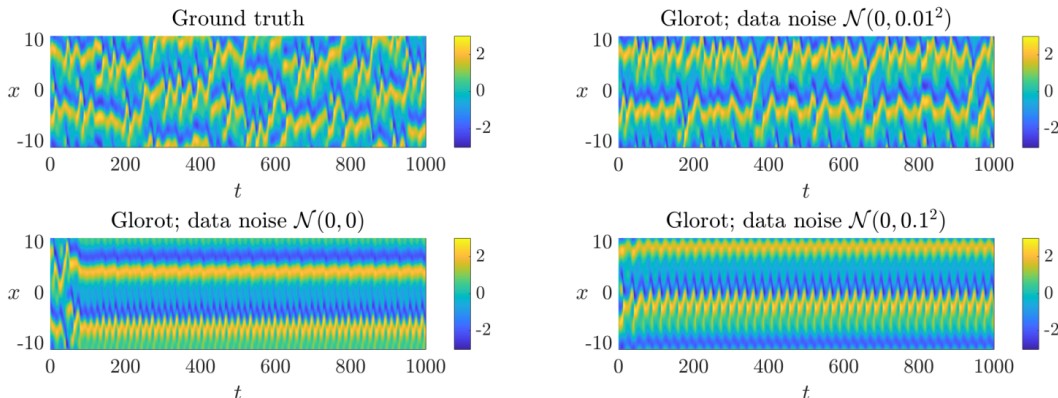

Figure 2: Trajectories produced by various discrete-time dynamical models starting from an initial condition in the test data. Shown are the ground-truth trajectory and trajectories produced by neural-network models with weights initialized using the Glorot normal initializer with varying degrees of noise added to the training data.

of the Kuramoto-Sivashinsky system, a nonlinear partial differential equation representative of the dissipative physical systems of interest.

The data come from a numerical simulation of the Kuramoto-Sivashinsky equation,

$$\frac{\partial v}{\partial t} = -v\frac{\partial v}{\partial x} - \frac{\partial^2 v}{\partial x^2} - \frac{\partial^4 v}{\partial x^4}, \qquad x \in [-L/2, L/2], \tag{26}$$

with periodicity boundary conditions. We set $L = 22$, which produces chaotic dynamics. The numerical simulation was performed using a pseudo-spectral method on a uniform grid with 64 points; as a result, the data have a dimension of $n = 64$. Time advancement was performed using exponential time differencing (Kassam & Trefethen, 2005) with a time step of $\Delta t = 0.25$. Details about the data and neural network architecture and training are available in Appendix C.

For discrete-time dynamical models, all of our models produce stable dynamics when the weights are initialized using the Glorot normal initializer (Figure 2). Accordingly, we did not test other weight initialization schemes. There are two likely culprits that lead to stable dynamics. The first is that the dimension and rank of the data may be sufficiently large that the eigenvalues of matrices produced using the Glorot normal initializer lie inside the unit circle. The second is that the presence of multiple layers in the neural networks leads to implicit regularization, which drives the neural networks towards low-rank solutions (Gunasekar et al., 2017; Arora et al., 2019). Explicit dimensionality reduction has previously been found to stabilize the learned dynamics (Linot & Graham, 2022; Chen et al., 2022; Floryan & Graham, 2022). A theoretical analysis of how the presence of multiple layers affects the learning dynamics is a promising avenue of future research.

The effects of training data noise are also investigated in Figure 2. For models trained on noise-free data, approximately half of them led to periodic dynamics, whereas the ground truth dynamics are chaotic. The addition of a small amount of noise to the training data rectified this behavior, producing chaotic dynamics. Adding a greater amount of noise to the training data led to periodic dynamics as well, and adding an even greater amount of noise led to stable fixed points (not shown). This is consistent with the stabilizing effect of noise that was noted in Section 2.

For continuous-time models, we expect them to be less stable since the theory in Section 3 shows that a greater fraction of eigenvalues are unstable in the linear case. This is indeed the case, as seen in Figure 3. With noise-free training data, Glorot normal weight initialization leads to unstable dynamics. Specifically, it is the low-energy high wavenumbers that are unstable, as the theory predicts. The addition of noise to the training data stabilizes the dynamics, though the energy in the large wavenumbers is still greater than in the ground truth dynamics. Adding a greater amount of noise leads to stable fixed points (not shown). We also tested the weight initialization scheme based on Gershgorin's circle theorem. Although it reduced the energy of large wavenumbers, it

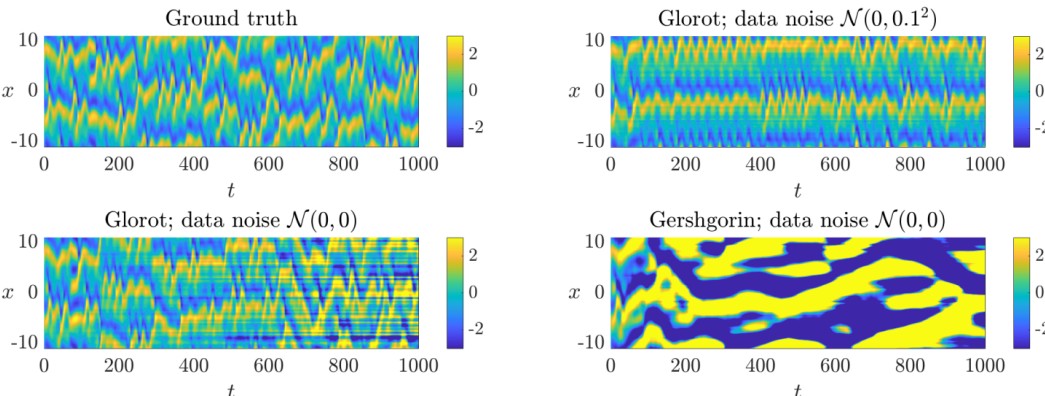

Figure 3: Trajectories produced by various continuous-time dynamical models starting from an initial condition in the test data. Shown are the ground-truth trajectory and trajectories produced by neural-network models with weights initialized using different initializers with varying degrees of noise added to the training data.

led to incorrect dynamics for the small wavenumbers. Investigating the behavior produced by this initialization scheme for neural networks with multiple layers is a future research direction.

## 5 DISCUSSION

Despite the importance of the long-term stability of learned physics simulators, theoretical insight into this issue is conspicuously missing. Our findings for linear dynamical systems constitute an important step towards a general theory, buoyed by nonlinear analogs with empirical support.

In the linear case, we showed that it is more difficult to learn low-energy dynamics due to the associated slower rates of convergence in the learning dynamics, and that learning is also more difficult in high-dimensional systems. This seems to hold for nonlinear systems as well, for which it has been empirically observed that it is difficult to learn the dynamics of high wavenumbers in physical systems, which have characteristically low energy (Chattopadhyay & Hassanzadeh, 2023; Lippe et al., 2024). Although the difficulty to learn the dynamics of high wavenumbers has previously been attributed to the spectral bias of neural networks (Chattopadhyay & Hassanzadeh, 2023; Xu et al., 2019; Rahaman et al., 2019), recent work on multi-stage neural networks supports that non-uniform distribution of energy and spectral bias both contribute to slow convergence (Wang & Lai, 2024). The non-uniformity of energy could be addressed by first transforming the variables to a space in which the data are distributed isotropically, then learning the dynamics of the transformed variables.

We then showed that dynamics off of the data subspace cannot be learned, and what is learned in the complement of the data subspace depends on the weight initialization. This obviously holds for nonlinear systems, where the dynamics off of the data submanifold cannot be learned. What is less clear is whether a non-trivial weight initialization scheme can be designed so that the default dynamics are stable, as we have done here for linear dynamical systems. In the linear case, an alternative is to project the system onto the data subspace. In the nonlinear case, manifold learning methods can be used to project the system onto the data submanifold; doing so has, indeed, been found to stabilize the learned dynamics (Linot & Graham, 2022; Chen et al., 2022; Floryan & Graham, 2022). Another alternative is to add global damping to the system (Vlachas et al., 2018; Linot & Graham, 2022; Linot et al., 2023), though it is unclear how strong it should be.

Finally, we noted the empirical success of noise injection as a stabilizer when learning nonlinear dynamical systems, and showed why it works when learning linear dynamical systems. Adding noise to the data adds damping to the learning dynamics. Damping is a generic mechanism that is likely to extend to the learning dynamics of nonlinear systems, as our numerical experiments showed. Furthermore, we showed that there is a tradeoff between the stabilization and bias that noise

creates, and this tradeoff can be seen in empirical results for nonlinear systems (Sanchez-Gonzalez et al., 2020).

The agreement with empirical results for nonlinear systems suggests that we have identified the correct mechanisms. However, our suggested mitigative strategies do not fare as well for deep nonlinear neural networks as they do for linear single-layer neural networks. Even for linear neural networks, the addition of layers leads to different learning dynamics (Saxe et al., 2014; 2019). Accordingly, the logical next step in the analysis of the learning dynamics of neural networks tasked with emulating dynamical systems is to analyze how increasing the depth of a linear neural network affects the learned dynamics. We believe this direction holds much promise, as it has provided much insight for static learning problems.

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

## A    DISTRIBUTION OF EIGENVALUES FOR DIFFERENT WEIGHT INITIALIZATION SCHEMES

Figure 4 shows a histogram of eigenvalues of an $n \times n$ matrix whose entries are generated using the Glorot normal initializer. For finite $n$, the eigenvalues can lie outside the unit circle.

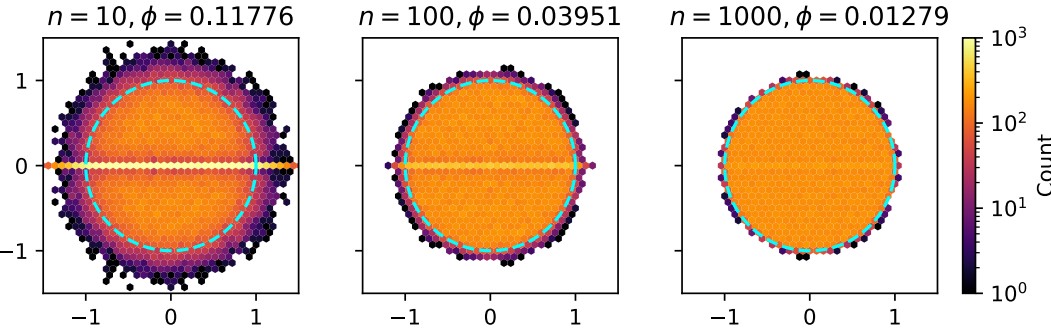

Figure 4: Histogram of eigenvalues of an $n \times n$ matrix whose entries are generated using the Glorot normal initializer. $10^5/n$ realizations of the random matrix were used. The unit circle is drawn with a dashed cyan line. $\phi$ gives the fraction of eigenvalues outside of the unit circle. The Glorot uniform initializer produces nearly identical histograms.

Figure 5 shows a histogram of eigenvalues of an $n \times n$ matrix whose entries are generated using the initialization scheme based on Gerhsgorin's circle theorem described in Section 2. The eigenvalues are guaranteed to lie inside the unit circle.

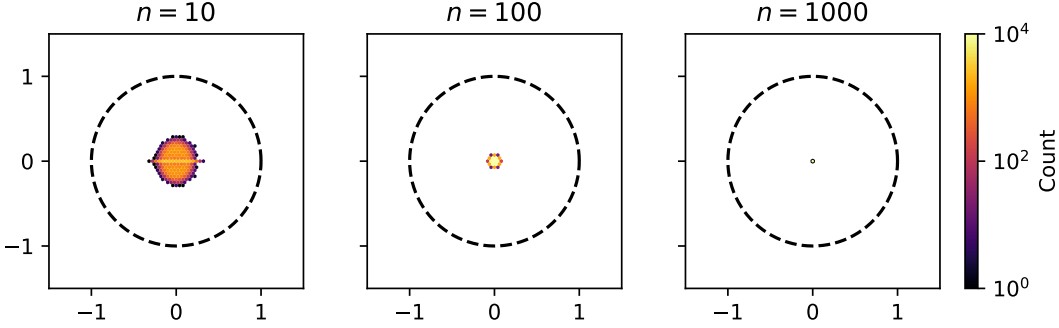

Figure 5: Histogram of eigenvalues of an $n \times n$ matrix whose entries are generated using the initializer based on Gershgorin's circle theorem. $10^5/n$ realizations of the random matrix were used. The unit circle is drawn with a dashed black line.

## B    DISCRETE-TIME DYNAMICAL SYSTEM IN FIGURE 1

The three-dimensional discrete-time dynamical system corresponding to Figure 1 is $x_{i+1} = Ax_i$, where

$$A = \begin{bmatrix} 0.5 & 0 & 0 \\ 0 & -0.5 & 0 \\ 0 & 0 & -0.5 \end{bmatrix}. \tag{27}$$

100 snapshot pairs were generated by evolving 100 random initial conditions with entries that are $\mathcal{O}(1)$ one step into the future. The third component of every initial condition was set equal to zero; therefore, the third components of all vectors in the dataset were zero. As a result, the dynamics of the third component cannot be learned from the dataset, and we refer to it as the unlearnable direction or mode. Note that the true dynamics in this direction are stable.

For the noisy dataset, the entries of $N_x$ and $N_y$ were drawn from $\mathcal{N}(0, 0.05^2)$.

## C  DATA AND NEURAL NETWORKS FOR THE KURAMOTO-SIVASHINSKY EXAMPLE

The training data consist of $m = 40000$ snapshot pairs taken from one long numerical simulation. The snapshots are 0.25 time units apart. The training data were collected after the system settled onto its attractor.

The results presented in the main text used the test data, which consist of 10000 snapshot pairs taken from a separate numerical simulation with a different initial condition. The test data were collected after the system settled onto its attractor. We have verified that the training and test data cover the attractor by inspecting the joint PDF of energy production and dissipation (Linot & Graham, 2020).

All neural networks were fully connected and had shapes $64 : 200 : 200 : 200 : 64$. Each layer had a sigmoidal activation function except the last layer, which had no activation function. Neural networks were trained using mini-batch gradient descent. The mini-batches had 2000 snapshot pairs. The Adam optimizer with a learning rate of $10^{-3}$ was used for training, which was conducted for 300 epochs. For continuous-time models, neural ODEs were used (Chen et al., 2018).

