# OpenReview forum: "Dynamics of learning when learning dynamics using neural networks"
_ICLR.cc/2026/Conference — Submitted to ICLR 2026_

### Official Review · Reviewer_XZ2m · 2025-10-30

**Soundness:** 2
**Presentation:** 3
**Contribution:** 2
**Rating:** 4
**Confidence:** 4

**Summary:**

Taking continuous and discrete linear systems as a case study, this paper analyzes the dynamics of the parameter matrix under Mean Square Error (MSE) and gradient descent, and derives three conclusions.

**Strengths:**

This paper is grounded in rigorous mathematical derivations and provides practical methodological guidance based on the theoretical results.

**Weaknesses:**

1. The model in this paper is overly simplistic (it employs a linear system).
2. The experiments are overly simplistic. Moreover, they lack comparative tests on real-world systems or public datasets. For instance, an ablation study on the dataset with and without added noise could have been designed.
3. Linear systems have analytical solutions, allowing gradients to be expressed explicitly. In contrast, obtaining gradients for nonlinear systems is considerably more difficult (often requiring numerical ODE solutions). This fundamental difference raises doubts about whether your analysis remains applicable in nonlinear cases.

**Questions:**

Can a similar theoretical analysis be conducted on nonlinear systems?

---

### Official Review · Reviewer_bM3b · 2025-11-01

**Soundness:** 2
**Presentation:** 3
**Contribution:** 3
**Rating:** 4
**Confidence:** 3

**Summary:**

This paper investigates why neural networks trained to model physical dynamical systems often produce unphysical or unstable long-term dynamics. The authors analyze the learning dynamics of gradient descent for linear dynamical systems using single-layer linear neural networks. Through analytical derivations, the key findings are: (1) uneven convergence rates tied to data energy distribution, with zero-energy directions being unlearnable; (2) dependence on weight initialization in unlearnable directions, where standard schemes (e.g., Glorot) can produce unstable eigenvalues; (3) Noise adds damping to the learning dynamics, but introducing a bias in the learnable modes. They extend the analysis to continuous-time dynamics and discuss qualitative connections to nonlinear systems.

**Strengths:**

- The paper's perspective is insightful. It shifts the blame for instability away from neural network expressivity and towards the learning dynamics of gradient descent. This analysis of the interplay between initialization, unlearnable subspaces, and gradient flow is a novel contribution.
- The paper is well-written. The theoretical analysis of the linear case is rigorous and mathematically sound.

**Weaknesses:**

1. The paper's claims are derived from a linear model ($\hat{A}x$) consisting of a linear single-layer neural network. The extension to deep, non-linear systems—which is where the actual problem lies—remains purely speculative. Although the authors claim this is an important starting point, it is not obvious that the learning dynamics of a deep network in "off-manifold" directions would simply "retain initialization" in the same way. It also remains unclear how to translate this Gershgorin-based principle into a practical deep-network initialization scheme.
2. A major weakness is the lack of empirical validation. The paper motivates its claims with real-world problems such as weather, but provides no experiments on simple chaotic systems (e.g., Lorenz-63, Kuramoto-Sivashinsky) to demonstrate that the identified mechanisms (initialization, adding noise) actually hold, or that the proposed mitigations (Gershgorin initialization) provide any benefit in the non-linear case.

**Questions:**

1. How do your proposed remedies (stable initialization, noise) compare to other common techniques used to stabilize NN dynamics, such as spectral normalization, orthogonal initialization/constraints, or direct eigenvalue regularization?
2. The entire analysis is based on continuous-time gradient flow. How do you expect your conclusions (especially regarding the "damping" effect of noise) to change when using practical mini-batch SGD or Adam? Could the optimizer's intrinsic noise provide a similar stabilization effect, potentially making explicit data noise redundant?

---

### Official Review · Reviewer_8Rr5 · 2025-11-01

**Soundness:** 1
**Presentation:** 2
**Contribution:** 1
**Rating:** 2
**Confidence:** 4

**Summary:**

This paper analyzes the dynamics of neural networks that are trained on time series data from dynamical systems. The neural network models the authors consider is a single layer linear network, and the case of learning discrete maps and continuous-time systems are treated separately. For the discrete time case, the authors conclude that the covariance of the dataset directly influences the learning dynamics, and that one must initialize the weight matrix so that its eigenvalues lie in the unit circle. On this front, the authors provide a new initialization scheme based on the Gershgorin circle theorem. Additionally considering the case of noisy data, the authors find that the signal-to-noise is directly related to the bias introduced in the final learned dynamics. For the continuous time case, the authors find similar results with slight differences due to the residual network-like computation structure of the problem. Namely, they find that the convergence rate of the dynamics dependent on the square of the data sampling period $\Delta t$ and that the eigenvalues of the weight matrix must have their real parts be positive.

**Strengths:**

1.	From analyzing a simplified version of the dynamical system learning problem, the authors provide some insights about the training process

2. The proposed weight initialization scheme seems interesting and serves as potentially interesting future research directions

**Weaknesses:**

1.	The neural network model considered in this paper is too simple. The authors call it a single layer linear neural network, but this is just linear regression performed with gradient descent. The authors do not provide any argument or experiments that justify that their linear regression setting-based conclusions are actually valid for nonlinear multi-layered neural networks. Therefore, it is difficult to gauge how much of the paper's conclusion are actually relevant to conventional neural network training.

2. Related to the previous point, there is almost no experiments that back up the predictions made in this paper. While the authors provide an initialization scheme, it is not actually used for model training to verify its efficacy. The convergence rate predictions or the noise-induced bias in the learned dynamics are not experimentally checked either. Figure 1 and 2 are just numerical confirmations of the Gershgorin circle theorem and Figure 3 is difficult to understand, since almost no details regarding the computation performed is provided.

**Questions:**

1.	Can the authors provide a justification why their linear single-layer neural network is a good model for neural network training? Can the authors give at least a qualitative argument of what would happen if the network is multi-layered and/or has nonlinearities?


2. The authors argue for the discrete case, the eigenvalues of the weight matrix must lie inside the unit circle. I am unsure if this argument directly follows from the solution of the learning dynamics in equation 8. Can the authors provide clarification? Are the authors referring to the fact that if the weight matrix has positive eigenvalues, the iterated predictions of the model will diverge?

3. What was done exactly for Figure 3? What was the 3D discrete-time system used? How was the model trained? What are the actual ground truth eigenvalues of the system?

4. Can the authors provide additional experiments that confirm their predictions made throughout the paper?

---

### Official Review · Reviewer_TWAp · 2025-11-03

**Soundness:** 4
**Presentation:** 4
**Contribution:** 3
**Rating:** 8
**Confidence:** 4

**Summary:**

In this paper, the authors study the problem of learning the governing law of a dynamical system under the constraint of linearity. Although this constraint is not typically used in modern deep learning models, the resulting theoretical framework provides rich explanations and guidance regarding optimization dynamics. The paper is also highly readable.

The paper is divided into two main parts: learning discrete dynamical systems and learning continuous dynamical systems.

The main contributions focus primarily on the discrete case, which is where this summary will focus the most. In the discrete case, the authors model the system: $x_{t+1} = f(x_t)$, where $f$ is a linear operator (i.e., $f(x) = Ax$). A neural network models this system with a single linear layer.

The optimal solution, $\hat{A}$ (the estimated linear layer weights), can be directly written as a linear combination involving the standard Moore-Penrose inverse of the data matrix $X$ and the label matrix $Y$. However, since neural networks are optimized using a gradient descent framework, it is straightforward to derive the gradient of the $L_2$ loss function. In the limit of a small learning rate, the optimization problem is described by continuous-time gradient flow dynamics, using pseudo-time (optimization time). Next, the authors perform a coordinate change using the left-singular vectors of the data matrix $X$ and re-write the gradient flow ODE. Since this is still a linear ODE, the analytical solution is easily obtainable, revealing three key components: the true dynamics operator, the initial NN weights, and an exponential function of the data covariance matrix (after the coordinate change).

The analytical solution for the neural network weights in pseudo-time has three important implications:

Columns of estimated weights corresponding to the non-zero singular values converge to the columns of the true dynamic operator, while the columns corresponding to the zero singular values retain their initial, randomly assigned values.

The rate of convergence of the linear neural network is linked to the singular values of the data matrix and the dimensionality of the problem.

Initialization impacts the learned dynamics in the "unlearnable" directions (zero singular values) and in directions where early stopping is applied.

In the case of noisy data, the authors repeat the analysis by evaluating the gradient flow with additional random matrices. In this setting and in the limit of infinite pseudo-time, the authors show that the learned columns of the dynamics include a bias term connected to the signal-to-noise ratio, and zero-singular columns are mapped to 0 (instead of retaining the initialization values).

The authors test these main findings in Figure 3, where they control the unlearnable mode, high and low singular values modes, and demonstrate how noise acts as a stabilizer.

In the continuous case, the authors again convert the problem to the previously described form by applying two approximations: i) Taylor/Euler forward expansion of the matrix exponential operator, and ii) again taking the limit of infinite pseudo-time. The main difference in the continuous case is that Glorot initialization leads to more frequent instability regimes. The authors suggest using an initialization based on Greshorin’s circle theorem to obtain a strict bound on the spectral values, which guarantees stability. Finally, the authors derive an expression with two main terms: a multiplicative bias correction factor (connected to the noise) and an additive term related to the $1/\Delta t$ solver discretization error.

**Strengths:**

- Simple but effective framework for explaining learning dynamics of linear neural networks for linear dynamics with Xavier initialization
- Strong theoretical reasoning

**Weaknesses:**

- Very limited experimental confirmations of the theoretical contributions. Practically only figure 3 is experimental verification
- Not well explained how and under which assumptions equation 9 is formulated. I can understand the logic w.r.t. invariant manifolds, but this part need more proper formulation for ML community
- No experimental confirmations for continuous dynamics
- No experimental checks how could one go from linear-type assumptions to the non-linear type assumption

**Questions:**

- Formulate equations 9 and 10 with invariant assumptions for dynamical systems with more proper formulation and not just hand-waving arguments from other papers
- Connect the main findings to learning from static data not dynamics, one can still apply singular value analysis?
- How could one make the next step and show how to proceed in the case of nonlinear dynamics at least with experimental verifications?

---

### Meta-Review · Area_Chair_HgP1 · 2025-12-15

**Summary:**

This paper investigates the origins of unphysical instabilities in neural networks modeling dynamical systems by analyzing the gradient descent learning dynamics of linear single-layer networks.

While the proposed analytical framework can be useful in some cases, the biggest concern is the gap between the theoretical model and the target application, which reviewers felt remain speculative and unproven. The numerical experiments that came with the original manuscript are inadequate by consensus. Authors did not provide rebuttal or experimental during the discussion period.

The recommendation is rejection.

**Reviewer Concerns:**

Authors did not provide rebuttals.

**Reviewer Scores:**

Reviewer scores would not have changed because authors did not provide rebuttals.

---

### Decision · Program_Chairs · 2026-01-26

Reject